# An online communication skills training program for nursing students: A quasi-experimental study

**Jeongwoon Yang**[1], **Sungjae Kim**[2]*

**1** Department of Nursing Science, Kyungbok University, Namyangju, Korea, **2** College of Nursing and the Research Institute of Nursing Science, Seoul National University, Seoul, South Korea

* sungjae@snu.ac.kr

## Abstract

In South Korea, in 2019, approximately 45.5% of newly-graduated nurses quit their jobs within one year of employment. To better understand the adjustment to nursing practice upon graduation, we developed an online communication skills training program based on nonviolent communication and evaluated its effectiveness. A quasi-experimental design was adopted. The sample included 28 participants in the experimental group and 27 in the control group after one participant in the control group dropped out. The participants were fourth-year nursing students at the K and S University in Gyeonggi Province, South Korea, with some clinical training in a hospital setting. Data were analyzed using the $\chi^2$ test, Fisher's exact test, and independent t-test. Participants' empathy, communication skills, anger, and self-efficacy were assessed before and after the training, as well as across the two groups. The experimental group showed significantly higher levels of empathy, communication skills, and self-efficacy compared to the control group after the program completion. However, there were no significant differences in anger. This study suggests the effectiveness of the online version of the nonviolent communication training. Therefore, providing this program to nursing students scheduled for graduation may help retain newly-graduated nurses.

## Introduction

As of 2019, approximately 20,000 nursing students graduated from the 203 nursing schools in South Korea [1]. After completing a minimum of 130 theoretical courses and 1,000 clinical hours to be equipped with the competency required of nurses in clinical practice, those who pass the licensure exam become registered nurses [1].

Approximately 45.5% of the newly-graduated nurses in Korea quit their jobs within one year of employment [2], many of whom leave the profession entirely. Thus, Korea has a relatively low proportion of active nurses among the Organization for Economic Co-operation and Development (OECD) countries, despite having the highest proportion of newly-graduated nurses [3]. In Korea, it is time-consuming for a newly-graduated nurse to exhibit at least

**Data Availability Statement:** All data generated or analysed during this study are included in the Supporting information files.

**Competing interests:** The authors have declared that no competing interests exist.

90% proficiency as compared to that of a skilled nurse; further, on an average, over 25% of the annual cost-to-company is spent on training newly-graduated nurses [4]. Hence, the resignation of nurses within one year of employment is inefficient not only for nurses themselves, but also for the organization.

Premature resignation of newly-graduated nurses has been attributed to rotating shifts, including night shifts [5], stress from inexperience in nursing-related tasks [6], workload and burnout due to understaffing [7], and inefficient communication with fellow nurses [6, 8]. Communication problems significantly contribute to interpersonal workplace conflict [9] and cause low work efficiency and job satisfaction [10, 11]. Newly-graduated nurses, when subjected to harsh criticism, become pessimistic about their competency and feel alienated from their organization [12]. This, in turn, markedly undermines their efficiency and confidence in nursing work [9], ultimately forcing them to quit their jobs [10]. Further, approximately 60–70% of nurses in Korea and abroad have reported to have experienced verbal violence in the hospital [9, 13], which highlights the gravity of such communication problems.

Newly-graduated nurses with low levels of empathy fail to engage in effective interaction and communication with their clinical supervisor, which increases their job stress and turnover intention [10]. Further, newly-graduated nurses who are not able to engage in respectful conversations with colleagues do not develop a sense of belonging, and experience elevated job stress [14]. These findings suggest that the communication difficulties experienced by newly-graduated nurses in their workplace are associated with their communication skills. Hence, fostering effective communication skills in newly-graduated nurses would help prevent premature resignation.

Studies have reported that nurses with effective communication skills show high levels of empathy and self-efficacy [9, 15]. That is, nurses adept at effective communication can engage in conversations with fellow nurses and patients using empathy. In addition, effective communication skills enhance work efficiency [16] and improve self-efficacy, thereby increasing job satisfaction [14]. In hospital settings, wherein nurses work on rotating shifts, effective communication among nurses is essential for continuity in patient care [17]. Moreover, because nursing work often involves highly stressful situations due to the urgency and gravity of dealing with human lives [18], lack of communication or minor communication errors may lead to medical malpractice [19]. Preceptors who train newly-graduated nurses often use a strict tone with them due to the risks and possibility of errors in patient care [20]; newly-graduated nurses may take such a tone personally [20] and react with feelings of anger. Considering such work situations and conditions, training in communication skills as part of undergraduate nursing education would help newly-graduated nurses improve their empathy, self-efficacy, and anger management ability [21]. Further, better communication skills could also help prevent premature turnover [22].

Communication skills can be improved through practice and training [23]. Effective communication skills involve clearly delivering one's opinion and accepting the other person's opinion [24], and nonviolent communication (NVC) encompasses these factors. Nonviolent communication, also referred to as compassionate communication, is a conversation model developed by Marshall Rosenberg based on Mahatma Gandhi's values and Carl Rogers' therapeutic principles [25]. It comprises two key factors: expressing honestly and listening empathetically [25]. It is a conversation technique based on compassion toward others, which also helps one engage in introspection and build relationships with others [25]. Therefore, by learning and practicing NVC, we reason that the nurses would be able to empathize with their fellow nurses, understand the meaning of their words and their intentions, and appropriately and honestly express their own emotions and needs in a clinical setting.

Currently available modules for communication training and education for nursing students focus on therapeutic communication when interacting with patients [26]. Such programs rarely train nursing students on self-expression during conversations with colleagues. To the best of our knowledge, there are no online NVC training courses that are specific to undergraduate nursing students or nurses; however, online NVC courses for health professionals in general and an offline course for nurses do exist [27]. Communication courses are generally lower division courses targeting first- or second-year students who have not begun their clinical training [28]. Therefore, it is helpful for fourth-grade students who are about to graduate to receive communication training once again since their skills are limited in that they may not necessarily translate their service to the context of understanding the reality of clinical nursing settings. However, due to the global pandemic (COVID-19), online education began in the first semester for the cohort starting in 2020 [29]. Thus, this study aimed to develop and evaluate the feasibility of an NVC-based online communication skills training program (NVC-CST program) to address the needs of nursing students. The NVC-CST program that we have developed focuses on fostering students' empathy, self-efficacy, appropriate anger expression, and communication skills.

## Materials and methods

### Design

The present study used a quasi-experimental pretest-posttest design. Volunteer participants were randomly assigned into the experimental group or control group using the Research Randomizer computer program (https://www.randomizer.org). The effectiveness of the online NVC-CST program was also tested.

### Participants

Fifty-six nursing students at the K and S University in Gyeonggi Province, South Korea, were enrolled in the study. The inclusion criteria were (a) fourth-year nursing students, and (b) having experienced clinical training at a hospital. The exclusion criteria were (a) minors under the age of 18 (as per Article 2 of the Child Welfare Act of the Republic of Korea) and (b) currently being on a leave of absence.

The sample size to achieve the study objectives was determined using the G*Power Program (version 3.1); the minimum sample size was calculated to be 42, with 21 participants in the experimental group and 21 in the control group (effect size = 0.80, significance = .05, power = .80, two-tailed test). Considering a potential 30% withdrawal rate, 56 students were enrolled. The number of groups (2) and the total number of people (56) for the groups were entered into the Research Randomizer program. Subsequently, the participants were assigned to the experimental and control groups based on the numbers assigned to them by the program. One participant from the control group dropped out (due to personal reasons, even after having attended the program twice), resulting in 28 participants in the experimental group and 27 in the control group. The effect size was established with reference to 0.87, as reported in a Korean study that analyzed NVC-based interventions [24].

### Procedure and data collection

The public institutional review board (IRB No. P01-202010-12-001) approved this study, and data were collected from October 5, 2020 to November 15, 2020. Additionally, the study protocol was in accordance with the Code of Ethics of the World Medical Association (Declaration of Helsinki). Students were informed about the nature of the study, following which they

provided signed consent to participate, prior to enrollment. Both the experimental and control groups completed the same questionnaire both before and after the intervention. The experimental group was instructed to access an online platform to participate in the NVC-CST program twice a week. Due to the pandemic, both sets of participants had been taking online classes without physically attending school; therefore, there was no contamination between the experimental and control groups. To prevent the diffusion effect, the duration of access to each 30-minute session was restricted to 3 hours. We also monitored the number of participants on the online platform to confirm that participants in the experimental group participated regularly. No communication program was provided to the control group. However, upon completion of the NVC-CST program, the control group students were allowed access to the program upon request.

## The Nonviolent Communication-based online Communication Skills Training (NVC-CST) program

The NVC-CST program aims to help students acquire the skills of expressing their emotions and situations, empathizing with others, and accepting others during conversations, by enabling the students to practice communication skills learned online.

Specifically, this program entails educating participants about the fundamental principles of NVC and helping them practice these skills by applying the principles while communicating with fellow nurses. Various communication case studies were developed with reference to qualitative studies that examined the clinical experiences of newly-graduated nurses in the Korean cultural context [12, 30]. The facilitator of the program was a nursing professor with 15 years of clinical nursing practice and experience in training various groups on NVC. The program utilized cognitive and behavioral approaches, including lectures, conversation demonstrations, assignments, and testimonies. A lecture was provided on each session's topic. Moreover, demonstrations of both ineffective communication and NVC-based communication were provided. Students were given an assignment to apply the topic of each session to daily life, and they were instructed to write a testimony after each session, except the introductory one.

The NVC-CST program involved four stages, comprising eight sessions overall (Table 1). Session 1 comprised an introduction to the NVC-CST program and an explanation of the objectives, with a focus on building a trusting relationship between the facilitator and the participants. Session 2 covered the four principles of NVC; the focus was on distinguishing between observations and feelings and understanding needs and requests. Students were assisted in understanding these four principles of NVC using a case study of a nurse from the period of his/her graduation to the achievement of a senior position. Session 3 focused on understanding that "feelings" and "needs" are connected, as well as understanding the positive energy of "needs". Further, an example of a conversation with a fellow nurse was provided to help students realize that others' needs are as important as one's own. Session 4 comprised various clinical practice cases to provide intensive training on empathetic listening and honest self-expression.

Session 5 comprised practices in correcting ineffective communication habits with the goal of fostering the competency to distinguish "observation" from "evaluation". In particular, examples of criticizing or disparaging conversations that commonly occur among nurses in clinical practice were used to show the difference between criticism and objective observation. Here, the focus was to distinguish between the two. Session 6 comprised practices intended to improve recognition of one's own and others' feelings and needs. In this session, instances of conversations in a clinical setting was presented to provide students an opportunity to

**Table 1. Nonviolent communication-based online communication skills training program.**

| Stage | Session | Contents | Time (minutes) |
|---|---|---|---|
| 1 Introduction | 1 | · Orientation | 30 |
| | | · Therapeutic alliance | |
| | | · Understanding the contents and objectives of the program | |
| 2 Understanding of the NVC principles | 2 | · Understanding of the four factors of the NVC | 30 |
| | | - observation, feelings, needs, request | |
| | 3 | · Connecting feelings and desires | 30 |
| | | · Staying in the energy of the needs | |
| | 4 | · Listening empathically | 30 |
| | | · Expressing honestly | |
| 3 Application of NVC in nursing practice | 5 | · Distinguishing between observation and evaluation | 30 |
| | | · Distinguishing between feelings and thoughts | |
| | 6 | · Awareness of own feelings and needs | 30 |
| | | · Awareness of others' feeling and needs | |
| | 7 | · Complete anger expression | 30 |
| | | · Stage of anger expression | |
| 4 Closing | 8 | · Expressing gratitude | 30 |
| | | · Obstacle of empathy | |
| | | · Self-reflection | |

NVC: nonviolent communication.

indirectly experience them. Session 7 involved learning about the appropriate ways to express anger and the steps involved in anger expression. Based on the NVC principle that views anger as an alarm for bodily and mental responses, this session trained students in appropriately expressing anger, instead of ignoring or suppressing it. For example, helping them objectively describe a situation in which they experienced anger. Session 8 comprised expressing gratitude and learning about the possible barriers to empathy. Various speech patterns that hinder empathy, such as analyzing and diagnosing, correcting, stopping the flow of emotion, sympathizing, and investigating, were described using examples of conversations among clinical nurses. In the latter part of Session 8, students were given time to briefly meditate and reflect on themselves.

## Measures

**Empathy.** The Empathy Quotient-Short Form (EQ-Short-K) [31], a Korean-adapted version of the Empathy Quotient-Short Form [32], was used to assess empathy. The EQ-Short-K has 11 items that are responded to using a 3-point Likert scale ranging from 0 (disagree) to 2 (strongly agree). Total scores can range from 0 to 22, with a higher score indicating higher levels of empathy. The reliability of this instrument, as measured by Cronbach's α, was .88 at the time of development [31] and .82 in the present study.

**Self-efficacy.** The New General Self-Efficacy Scale (NGSE) [33] was employed in the present study to assess self-efficacy; it has been used in a previous Korean study [34]. This instrument has eight items that are responded to using a 7-point Likert scale. The average score is the mean item response, ranging from 1–7, where a higher score indicates higher self-efficacy. Cronbach's α was .84 in the Korean study [34] and .91 in the current study.

**Anger.** The Korean State-Trait Anger Expression Inventory (STAXI-K) [35] is a Korean standardized version of Spielberger's [36] STAXI. There are 24 items that assess anger

expression across three subscales: Anger-in, Anger-out, and Anger Control. Items are responded to using a 4-point Likert scale. The anger score was calculated as follows: [(Anger-in score + Anger-out score)–Anger Control score + 16]. Total scores range from 0 to 72, with higher scores indicating higher levels of anger expression. Cronbach's α was .74 for Anger-in, .73 for Anger-out, and .81 for Anger Control in the Chon et al. [35] STAXI-K study; in the current study, these values were .68, .76, and .78, respectively.

**Communication skills.** Communication competencies were measured using the Global Interpersonal Communication Competency Scale (GICC-15) [37]. The GICC has 15 items that are responded to using a 5-point Likert scale and consists of three subscales: Relationship, Language, and Interpersonal Competency. The total score is the mean item response, with a range of 1–5, where a higher score indicates higher communication skills. Hur [37] reported a Cronbach's α = .72 at the time of development. Cronbach's α was .86 in the current study.

## Data analysis

Participant characteristics were analyzed using frequencies, percentages, means, and standard deviations. The normality of the baseline scores for both groups was tested using the Kolmogorov–Smirnov test, and baseline homogeneity was tested using the Chi-squared test, Fisher's exact test, and $t$-test. The differences between the two groups with respect to empathy, self-efficacy, anger, and communication skills based on the NVC-CST program were analyzed using independent sample $t$-tests. SPSS version 26.0 (IBM Corp., Armonk, NY, USA) was used for the analyses, and the significance level was set at $p < .05$.

## Results

Participants who participated in the NVC-CST were the experimental group, and those who did not participate were set as the control group. Both the experimental ($M_{age}$ = 24.4 years) and control groups ($M_{age}$ = 24.3 years) were fourth-year nursing students, of whom 11 (19.6%) were men and 44 (80.4%) were women. Forty-one students (75%) lived at home, six (10.7%) in a dorm, and eight (14.3%) lived alone. There were no statistically significant differences in the demographic characteristics between the two groups at baseline. Further, there were no statistically significant differences in empathy, anger, communication skills, or self-efficacy between the two groups at baseline (Table 2).

At posttest, the experimental group showed significant changes in empathy, self-efficacy, and communication skills, after the eight-session NVC-CST program, compared to the control group. However, there were no significant differences in the anger scores between the two groups (Table 3).

## Discussion

This study examined the effectiveness of an NVC-based online communication skills training program for fourth-year nursing students who were scheduled to graduate. Given that in-person classes or direct contact was not an option due to the COVID-19 pandemic, the program was designed to be administered online. An online program was anticipated to have limitations considering that NVC-based communication training largely utilizes demonstrations resembling the actual situation, role play, and group activities [38]. However, the results of this study indicate that an online program can be effective as well. Previous studies have suggested that the effectiveness of online education is influenced by the duration of the program, the lecturer's competence, and learners' motives [39]. Therefore, we speculated that the duration of the NVC-CST program in this study, competence of the facilitator, and study participants were appropriate for this purpose. Specifically, eight 30-minute sessions can be considered

**Table 2. Homogeneity of outcome variables.**

| Variable | | Experimental group (n = 28) | Control group (n = 27) | t | p |
|---|---|---|---|---|---|
| | | n (%) or Mean ± SD | | | |
| Age (years) | | 24.6 ± .37 | 24.3 ± .29 | .608 | .546 |
| Gender | Male | 6 (10.9) | 5 (9.1) | -.331 | .742 |
| | Female | 22 (40) | 22 (40) | | |
| Residence | Home | 20 (36.4) | 18 (32.7) | 1.100 | .276 |
| | Dorm | 2 (3.6) | 4 (7.2) | | |
| | Alone | 5 (9.1) | 3 (5.5) | | |
| Empathy | | 9.27 ± 3.13 | 9.21 ± 2.05 | .127 | .899 |
| Self-efficacy | | 4.82 ± 1.11 | 5.03 ± .73 | -.862 | .392 |
| Anger | | 32.68 ± 4.16 | 33.29 ± 3.49 | -.591 | .625 |
| Communication skills | | 3.79 ± .44 | 3.85 ± .35 | -.491 | .557 |

effective for online communication training [40]. Further, various approaches were utilized, such as providing lectures and demonstrations by a skilled expert and writing testimonies of students' experiences. These appear to have engaged the students and strengthened their learning motivation. Moreover, most participants were fourth-year students who were scheduled to become newly-graduated nurses within a year. We reason that they would have been motivated to acquire communication skills that would help them in clinical practice. Furthermore, this study's results confirmed that besides knowledge, skills can also be acquired via an online program, consistent with previous results that showed that clinical training can be performed online [41]. Moreover, watching videos online also fosters learning [42]. This implies that providing repetitive training by periodically offering the NVC-CST program online would help students make effective communication a habit.

The significantly higher posttest scores for empathy in the experimental group compared to the control group could be attributed to the fact that NVC, which is the fundamental principle of the NVC-CST program, is based on respect and the understanding of others [38, 43].

In light of past findings indicating that empathy has a positive effect on one's interpersonal competency [38, 44], and that empathy of newly-graduated nurses affects their interactions with their preceptors [10], nursing students who completed training through the NVC-CST program would be more likely to have improved interpersonal competency, and, therefore, have fewer difficulties when interacting with their preceptors as newly-graduated nurses. This is because the program provides for practicing conversations using the NVC principle of observation, where the other person's words and behaviors are accepted objectively without

**Table 3. Differences in the dependent variables between groups.**

| Variable | Time | Experimental group (n = 28) | Control group (n = 27) | t | p |
|---|---|---|---|---|---|
| | | Mean ± SD | | | |
| Empathy | Pre | 9.27 ± 3.13 | 9.21 ± 2.05 | | |
| | Post | 12.00 ± 3.13 | 11.11 ± 1.96 | 1.28 | .021 |
| Self-efficacy | Pre | 4.82 ± 1.11 | 5.03 ± 0.73 | | |
| | Post | 5.88 ± 0.76 | 5.27 ± 0.75 | 3.02 | .004 |
| Anger | Pre | 32.68 ± 4.16 | 33.29 ± 3.49 | | |
| | Post | 33.57 ± 3.74 | 34.89 ± 3.30 | -1.40 | .363 |
| Communication skills | Pre | 3.79 ± 0.44 | 3.85 ± 0.35 | | |
| | Post | 4.27 ± 0.37 | 4.06 ± 0.34 | 2.18 | .034 |

judgment or criticism; consequently, the person can carry on a conversation based on respect and consideration [25]. Furthermore, engaging in a mutually respectful, empathetic conversation by recognizing one's own needs in addition to others', and by not trying to assert one's own needs, would deepen the bond during a conversation [25].

Consistent with previous studies indicating that NVC training resulted in significantly improved communication skills [25, 27, 38], this study found that NVC training significantly improved the posttest scores on communication skills. Thus, communication skills can be improved with practice and training [24, 43], and an online NVC-CST program can be as effective as in-person training. Thus, this study's results support expanding the platform for delivering communication training programs. Furthermore, the take-home assignments and brief testimonies utilized in the NVC-CST program also probably contributed to increasing the effectiveness of the program. Though practicing communication skills through methods such as role play is effective, having students apply the learned skills when talking with their family and friends in their daily lives contributed to developing their communication skills. Additionally, instructing participants to write a short testimony of their experiences while completing the given assignments helped them recognize the positive changes in their conversations, which further strengthened their motivation for learning. In a previous study, engaging in NVC increased one's self-expression and understanding of others [23], which is consistent with the results of the present study.

Self-efficacy refers to confidence in resolving a difficult situation or a challenging task [23]. Our results are consistent with previous findings that hospital staff who underwent a program that integrated mindfulness and NVC showed marked improvements in self-efficacy [45]. Thus, acquiring effective communication skills through the NVC-CST program can increase a nursing student's sense of self-efficacy in dealing with various situations.

Next, the NVC-CST program did not affect the participants' anger expression, which is inconsistent with previous results showing that NVC positively changed the anger expression of inpatients receiving treatment for alcohol use disorder [23]. This may be attributable to differences in the study samples. Specifically, the present study utilized a student sample with no known anger expression issues, whereas the previous study employed individuals with an alcoholism problem who had issues with emotion regulation. Furthermore, the period during which this study was conducted may have affected the results. Aside from the possibility that the participants were not specifically placed in situations that could cause anger, it is also possible that the online nature of the NVC-CST program did not provide ecologically valid communication training by introducing anger-provoking situations. Anger management is not a skill like empathy or communication that can be improved with repeated training; instead, it requires an exploration of individual needs in the contexts of anger-provoking situations or people. Therefore, it would have been difficult to provide training specific to anger expression through an online program. Further, though the stages of anger expression were described in words, the details were perhaps not sufficient to vividly portray the reality, which would have undermined the effectiveness of the training. However, while levels of anger did not diminish, improvements in other skills such as empathy, communication skills, and self-efficacy may have also improved anger management. Designing a role play to practice NVC when nurses experience frustration and anger because their needs are not met in clinical settings could further enhance the effectiveness of the program.

## Conclusion

In this study, an online NVC-CST program was provided to fourth-year nursing students who were scheduled for graduation, to examine its effectiveness. The program improved students'

empathy, communication skills, and self-efficacy. Administering the program to newly-graduated nurses could improve their nursing performance. Communication skills can be improved through practice, confirming the program's effectiveness for nursing students scheduled for graduation. Repeated exposure to the NVC-CST program before graduation could help these students maintain good relations with their fellow nurses. This would be based on empathy and an understanding of needs in various communication situations, such as handovers, after being trained by their preceptors in a clinical practice. This could ultimately help newly-graduated nurses adjust to the clinical setting and prevent premature resignations. Notably, this study substantiates the effectiveness of an online education/training program; the NVC-CST program can be utilized in communication training for nursing students without time and place restrictions. However, the unidirectional format should be revised to implement measures to enhance students' concentration and participation. We also suggest advancing this program by utilizing virtual reality or online simulations, along with communication education and training scenarios for use with nursing students.

This study demonstrates what is likely to help new nurses adapt to their clinical practice. Successful clinical adaptation also allows nurses to keep their jobs. Although there are various factors related to the turnover of hospital nurses, the strength of this study is that it focuses on areas that can be improved.

It often takes long to improve organizational, environmental and cultural factors, and structural problems that contribute to higher turnover. This implies that turnover is influenced by various factors that are involved in a complex relationship. Nevertheless, from the perspective of a new nurse, NVC can be a passive strategy for helping them continue in their jobs and preventing interpersonal problems. This is because, through NVC, the ability to express one's needs and understand others' points of view can be improved, which are vital for appropriate workplace engagement.

However, this study has several limitations. The participants were sampled from nursing schools in a specific region; therefore, the findings may not be generalizable to nursing students in other schools. Additionally, it was not possible to directly verify whether this program had a positive effect on the turnover rate and whether the participants experienced improved interpersonal relationships in their actual clinical practice. Future research evaluating the effectiveness of the program should include follow-up assessments after graduation and in the first year of work. For this reason, we propose a follow-up study comparing NVC training among first-year nurses and nurses not trained in NVC.

## Supporting information

**S1 Data.**
(XLSX)

**S2 Data.**
(XLSX)

## Acknowledgments

We would like to thank our research assistant, Dr. Soonah Jung, who assisted us with data collection.

## Author Contributions

**Conceptualization:** Sungjae Kim.

**Data curation:** Jeongwoon Yang.

**Formal analysis:** Jeongwoon Yang.

**Investigation:** Jeongwoon Yang.

**Methodology:** Sungjae Kim.

**Project administration:** Jeongwoon Yang.

**Resources:** Jeongwoon Yang.

**Software:** Jeongwoon Yang.

**Supervision:** Sungjae Kim.

**Validation:** Sungjae Kim.

**Visualization:** Jeongwoon Yang.

**Writing – original draft:** Jeongwoon Yang.

**Writing – review & editing:** Sungjae Kim.

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
