## [Decision Letter · Decision Letter 0]

10 Dec 2021

PONE-D-21-33565An online communication skills training program for nursing students: A quasi-experimental studyPLOS ONE

Dear Dr. Kim,

Thank you for submitting your manuscript to PLOS ONE. After careful consideration, we feel that it has merit but does not fully meet PLOS ONE’s publication criteria as it currently stands. Therefore, we invite you to submit a revised version of the manuscript that addresses the points raised during the review process.

We look forward to receiving your revised manuscript.

Kind regards,

Prabhat Mittal, Ph.D.

Academic Editor

PLOS ONE

Journal Requirements:

Additional Editor Comments:

Refer to the following citations

Mittal, P., & Raghuvaran, S. (2021). Entrepreneurship education and employability skills: the mediating role of e-learning courses. Entrepreneurship Education, 4(2), 153–167. https://doi.org/10.1007/s41959-021-00048-6

Mittal, P. (2020). Impact of Digital Capabilities and Technology Skills on Effectiveness of Government in Public Services. In 2020 International Conference on Data Analytics for Business and Industry: Way Towards a Sustainable Economy, ICDABI 2020 (pp. 1–5). IEEE. https://doi.org/10.1109/ICDABI51230.2020.9325647

Chakraborty, P., Mittal, P., Gupta, M. S., Yadav, S., & Arora, A. (2021). Opinion of students on online education during the COVID-19 pandemic. Human Behavior and Emerging Technologies, 3(3), 357–365. https://doi.org/10.1002/hbe2.240

Yadav, S., Chakraborty, P., Mittal, P., & Arora, U. (2018). Children aged 6–24 months like to watch YouTube videos but could not learn anything from them. Acta Paediatrica, International Journal of Paediatrics, 107(8), 1461–1466. https://doi.org/10.1111/apa.14291

Yadav, S., Chakraborty, P., & Mittal, P. (2021). User Interface of a Drawing App for Children: Design and Effectiveness. In Advances in Intelligent Systems and Computing (Vol. 1165, pp. 53–61). https://doi.org/10.1007/978-981-15-5113-0_4

Reviewers' comments:

Reviewer's Responses to Questions

**Comments to the Author**

1. Is the manuscript technically sound, and do the data support the conclusions?

Reviewer #1: Partly

Reviewer #2: Yes

2. Has the statistical analysis been performed appropriately and rigorously? 

Reviewer #1: I Don't Know

Reviewer #2: Yes

3. Have the authors made all data underlying the findings in their manuscript fully available?

Reviewer #1: No

Reviewer #2: Yes

4. Is the manuscript presented in an intelligible fashion and written in standard English?

Reviewer #1: Yes

Reviewer #2: Yes

5. Review Comments to the Author

Reviewer #1: The manuscript discusses the impact of an online training program on attributes that affect interpersonal communication like empathy, self-efficacy, and anger.

I am finding it difficult to make a recommendation for this article as it does not seem to report some important pieces of information to the reader:

1. Please submit the list of questions asked as a part of the standard questionnaires used. Please submit literature showing the validity and generalizability of these psychometric/behavioral assessment tools preferably in relation to other tools, if any. This is important to gauge the relative merits of these tools. It is difficult to rely on the study results without solid literature on the validity and generalizability backing the methods section.

2. The data tables need to show distribution of each test score as we are dealing with a sample of less than 30 units per control and experiment. Inferences from small samples have a higher risk of getting swayed in one direction or the other.

3. The study does not give details of when the training was conducted and when the pre-post tests were done. This in the case of an intervention that is online and informative in nature is important as it determines the test results.

4. The authors also need to clarify how the pre-post surveys were done. Were they online too?

I am a bit worried about the ramifications that can be drawn from these results in the absence of the above listed information.

Reviewer #2: The authors highlighted an important issue in the nursing practices in South Korea. Appreciated. However I have few comments.

1. How did you select your sample size?

2. Provide few more lines about research randomizer and G power program to make it more comprehensible for the readers.

3. Who designed NVC-CST program. The authors mentioned the program facilitator but I found lack of any information about who designed the program and what is the foundation of that training program.

4. Authors mentioned some causes (including inefficient communication) of premature resignation of newly graduate nurses, however I am interested to know why authors focused only one reason.

Rest is OK from my side.

6. PLOS authors have the option to publish the peer review history of their article (what does this mean?). If published, this will include your full peer review and any attached files.

Reviewer #1: No

Reviewer #2: No

---

## [Author Response · Author response to Decision Letter 0]

27 Mar 2022

Response to reviewers

[Reviewer #1]

1. Please submit the list of questions asked as a part of the standard questionnaires used. Please submit literature showing the validity and generalizability of these psychometric/behavioral assessment tools preferably in relation to other tools, if any. This is important to gauge the relative merits of these tools. It is difficult to rely on the study results without solid literature on the validity and generalizability backing the methods section.

Response

Thank you for your detailed review. The assessment tools used in the study have been modified and used according to the culture of Korean students (Please refer below). The questionnaire was of an appropriate length and showed a good sense of self-efficacy. We received content effectiveness index (CVI) advice from two experts from the School of Nursing. The CVI score was found to be 0.8. 

2. The data tables need to show distribution of each test score as we are dealing with a sample of less than 30 units per control and experiment. Inferences from small samples have a higher risk of getting swayed in one direction or the other.

Response

Thank you for your detailed review. The normality of the baseline scores for both groups was tested using the Kolmogorov–Smirnov test, and baseline homogeneity was tested using the Chi-squared test, Fisher’s exact test, and t-test.

3. The study does not give details of when the training was conducted and when the pre-post tests were done. This in the case of an intervention that is online and informative in nature is important as it determines the test results.

Response

The pre-tests of the experimental and the control groups were conducted online on October 5, 2020, using Google questionnaire. The NVC program, in which only the experimental group participated, was conducted between October 5 and October 30, 2020. An online post-test using Google questionnaire was conducted for both experimental and control groups on October 30, 2020.

4. The authors also need to clarify how the pre-post surveys were done. Were they online too?

Response

Thank you for your review. The pre-post surveys were conducted online using Google forms.

[Reviewer #2]

1. How did you select your sample size?

Response

We referred to previous studies related to NVC (Yang, J., & Kim, S. (2020). Effects of a nonviolent communication-based training program for inpatient alcoholics in South Korea. Perspectives in Psychiatric Care).

The sample size to achieve the study objectives was determined using the G*Power 3.1 Program; the minimum sample size was calculated to be 42, with 21 in the experimental group and 21 in the control group (effect size 0.80, significance .05, power .80, two-tailed test). Considering a potential 30% withdrawal rate, 56 students were enrolled. One control group participant dropped out, resulting in 28 participants in the experimental group and 27 in the control group. The effect size was established to be 0.87 in a Korean study that analyzed NVC-based interventions (Han, M., & Lee, K. (2017). Effects of communication ability enhancement program for nursing students in Korea: A systematic review and meta-analysis. The Journal of Korean academic society of nursing education, 23(1), 15–26.).

2. Provide few more lines about research randomizer and G power program to make it more comprehensible for the readers.

Response

In this study, participants were assigned to experimental and control groups using the Research randomizer (https://www.randomizer.org/) program. The number of groups (two) was entered into the program, and the total number of people (56) for the group was entered. Subsequently, the participants were assigned to the experimental and control groups based on the numbers assigned to them by the program.

3. Who designed NVC-CST program. The authors mentioned the program facilitator but I found lack of any information about who designed the program and what is the foundation of that training program.

Response

NVC-CST was developed by the main researcher (Jeongwoon Yang). Also, NVC-CST is a program developed based on NVC (Nonviolent Communication), which was developed by Marshall Rosenberg. It is a specific conversational model that focuses on two components essential to building mature relationships: honest expression and empathic listening (Rosenberg, Marshall B. (1983). A Model for Nonviolent Communication. Philadelphia PA: New Society Publishers.).

4. Authors mentioned some causes (including inefficient communication) of premature resignation of newly graduate nurses, however I am interested to know why authors focused only one reason.

Response

I agree with your opinion that, there are various factors such as the working environment, wages, and communication conflicts in the turnover of new nurses. It is impossible to solve structural problems such as the working environment and wages with individual efforts. However, conflict resolution through the acquisition of communication skills is possible through individual efforts. Therefore, this study emphasized communication skill training.

---

## [Decision Letter · Decision Letter 1]

21 Apr 2022

An online communication skills training program for nursing students: A quasi-experimental study

PONE-D-21-33565R1

Dear Dr. Kim,

We’re pleased to inform you that your manuscript has been judged scientifically suitable for publication and will be formally accepted for publication once it meets all outstanding technical requirements.

Kind regards,

Sergio A. Useche, Ph.D.

Academic Editor

PLOS ONE

Additional Editor Comments (optional):

Reviewers' comments:

Reviewer's Responses to Questions

**Comments to the Author**

1. If the authors have adequately addressed your comments raised in a previous round of review and you feel that this manuscript is now acceptable for publication, you may indicate that here to bypass the “Comments to the Author” section, enter your conflict of interest statement in the “Confidential to Editor” section, and submit your "Accept" recommendation.

Reviewer #1: All comments have been addressed

Reviewer #2: All comments have been addressed

2. Is the manuscript technically sound, and do the data support the conclusions?

Reviewer #1: Partly

Reviewer #2: Yes

3. Has the statistical analysis been performed appropriately and rigorously? 

Reviewer #1: Yes

Reviewer #2: Yes

4. Have the authors made all data underlying the findings in their manuscript fully available?

Reviewer #1: Yes

Reviewer #2: Yes

5. Is the manuscript presented in an intelligible fashion and written in standard English?

Reviewer #1: Yes

Reviewer #2: Yes

6. Review Comments to the Author

Reviewer #1: please add a sentence or two detailing when the pre and post test were conducted. The dates of the intervention and the pre and post test will be helpful.

Reviewer #2: (No Response)

7. PLOS authors have the option to publish the peer review history of their article (what does this mean?). If published, this will include your full peer review and any attached files.

Reviewer #1: No

Reviewer #2: No

---

## [Editor Report · Acceptance letter]

25 Apr 2022

PONE-D-21-33565R1 

An online communication skills training program for nursing students: A quasi-experimental study 

Dear Dr. Kim:

I'm pleased to inform you that your manuscript has been deemed suitable for publication in PLOS ONE. Congratulations! Your manuscript is now with our production department. 

Kind regards, 

on behalf of

Dr. Sergio A. Useche 

Academic Editor

PLOS ONE